# Synthesis of Amidoxime Adsorbent by Radiation-Induced Grafting of Acrylonitrile/Acrylic Acid on Polyethylene Film and Its Application in Pb Removal

**DOI:** 10.3390/polym14153136

**Published:** 2022-08-01

**Authors:** Rania Fekry Khedr

**Affiliations:** 1Chemistry Department Al Leith, University College, Umm Al-Qura University, Mecca 24382, Saudi Arabia; rfkhedre@uqu.edu.sa; Tel.: +966-557384005; Fax: +966-177423778; 2Department of Siting and Environment, Nuclear and Radiological Safety Research Center (NRSRC), Egyptian Atomic Energy Authority (EAEA), Cairo 11762, Egypt

**Keywords:** graft copolymerization, radiation, LDPE, Pb^+2^ removal

## Abstract

In the aquatic environment, heavy metals such as lead ions Pb (II) are of particular importance. These are due to Pb (II) being toxic at concentrations over 0.01 mg/L, when taken continuously over an extended length of time. Organs including the heart, gut, and kidneys are seriously harmed by Pb (II) intoxication. The neurological, reproductive, and bone systems are also affected. The removal of Pb (II) from aquatic environments is, therefore, crucial. Low density Polyethylene (LDPE) is grafted by radiation with Acrylonitrile and acrylic acid (PE-g-AN/AAc) for the adsorption of Pb (II). Factors that control the grafting process for optimum conditions, such as the effect of solvents, the air atmosphere, inhibitors, comonomer concentration, and composition and irradiation dose, are studied to obtain a high grafting yield without homopolymer formation and a higher water uptake. The results showed that the addition of 2.5% by wt% ferric chloride salt effectively inhibits homoploymerization of a mixture of 30% methanol and 70% H_2_O used as a solvent in nitrogen. The highest graft yield obtained was 320% at a 25 kGy radiation dose with an 80/20 monomer composition and 60% comonomer concentration. The resulting composite films were characterized by XRD to analyze the dispersion properties of the material, SEM for the surface morphology, FTIR analysis for the functional groups, TGA, DSC for the thermal stability and elongation, and tensile strength for the mechanical properties. The uptake of Pb (II) from lead nitrate aqueous solution by (PE-g-AN/AAc) was observed under different conditions of the degree of grafting, contact time, metal ion concentration, and pH. The results obtained suggest LDPE-g-p (AN/AAc) as a superabsorbent for the Pb (II) ion’s removal from an aqueous solution.

## 1. Introduction

Many sectors, including metallurgical operations, welding, alloy making, the fertilizer industry, agricultural activities, tanneries, and electroplating and metal plating facilities, discharge heavy metal pollutants into the environment. The traces of heavy metals detected in the aqueous stream are very toxic to human health and the environment. As they are not biodegradable, heavy metals tend to accumulate in natural biological systems and can result in major health issues. Diverse industries and industrial units utilize lead for various uses and discharge huge amounts into the environment, including tanneries, paints and pigments, electroplating, metal processing, wood preservatives, textiles, the dye industry, steel fabrication, and canning [1]. Thus, it has become a difficult problem for researchers throughout the world to remove dangerous heavy metals from water bodies and industrial wastes. [2] Heavy metals may be removed from water environments using a variety of methods, including adsorption, ion-exchange, chemical precipitation, membrane filtration, electrochemical technology, and reverse osmosis [3,4,5,6,7,8,9,10]. However, most treatment technologies are economical. Nonetheless, adsorption is one of the more straightforward, simple to use, and inexpensive methods for removing heavy metals among the methods mentioned above.

For the removal of harmful inorganic components from industrial wastewater discharge, a variety of adsorbents including activated carbon, fly ash, sawdust, crab shell, coconut shell biomass, peat, chitosan polymer, natural zeolite, and silicates were examined. In compared to other traditional methods, adsorption has been the most employed due to its low cost, availability, ease of operation, and efficiency. Many different types of adsorbents have been researched for the removal of lead such as activated carbon, biomaterials, and nanomaterials [11,12,13,14,15].

Polyethylene (PE) was selected as the base polymer for heavy metal ion adsorption due to its excellent mechanical and thermal characteristics, as well as its low cost. In several investigations, single- or binary-monomer-grafted PE films have been examined. The radiation-induced grafted polymers may be applied to any kind of base polymer since they are chemically and mechanically stable [16,17,18,19,20].

In the current work, gamma radiation was used to create active sites on an inert polymer that are suitable for adding monomers and radicals to create grafts. This was done at an ambient temperature. This method is carried out using conventional chemical reactions. In addition, it is the cleanest and most adaptable grafting technique currently in use. The resulting films exhibit excellent swelling, thermal stability, and mechanical characteristics. The synthesis of (Acrylonitrile/Acrylic acid (AN/AAc)-g-low-density polyethylene (LDPE) were carried out by using a mixture of 30% methanol and 70% H_2_O as a solvent to provide a homogenous polymerization. The effect of various factors such as the comonomer concentration, comonomer composition, and irradiation dose were discussed. Furthermore, the present work explores the possibility of using LDPE-g-p (AN/AAc) to adsorb (Pb^+2^) from contaminated wastewater. The factors that influence the metal removal were studied, such as the effect of the grafting percentage, concentration of metal ion, contact time, and effect of pH.

## 2. Experimental

### 2.1. Materials and Reagents

1-(LDPE) films, with thickness of 25 μm, were provided by EL-Nasr Chemical Co., Egypt.2-Acrylic acid (AAc) of purity 99.9% (Merck, Darmstadt, Germany).3-Acrylo nitrile (AN) of purity 98.9% (Merck, Darmstadt, Germany).4-Ferric chloride, methanol, acetone was obtained from Merck, Darmstadt, Germany.5-Hydroxyl amine hydrochloride was supplied by Sigma Aldrich, St. Louis, MO, USA.6-Lead (II) acetate trihydrate (Merck, Darmstadt, Germany).

No further purification was done for all these materials, and they were used as received.

### 2.2. Preparation of Membranes (Grafting of AN/AAc onto the PE Films by Gamma Radiation)

Strips of LDPE were cut into “4 cm × 5 cm”, washed and weighed with acetone, dried in a vacuum oven at 50 °C, and weighed. A known weight of ferric chloride FeCl_3_ (2.5% by weight) acted as inhibitor that prevented homoploymerization. FeCl_3_ was dissolved in mixture of distilled water and methanol (with concentrations of 70 and 60% by weight). Different comonomer compositions of AN/AAC were added, depending on the required ratio of the comonomers concentration. Polymer strips were immersed in this mixture in test tubes and then degassed by bubbling of pure nitrogen gas for 5 min and subjected to Co^60^ Gamma irradiation. The membranes formed were washed several times with acetone and bidistilled water to remove residual inhibitors, monomers, and homopolymers of AN/AAc that had collected in the film. The films were weighed after being dried in a vacuum oven at 50–60 °C. Degree of grafting % was calculated as follows [21]:Degree of grafting %= [(W_1_ − W_0_)/W_0_] × 100
where W_1_ is the dry weight of grafted PE film and W_0_ is the dry weight of PE film.

### 2.3. Preparation of Cation Exchange Membrane (Amidoximation of Nitrile Group of AN/AAC-Grafted LDPE Film)

Figure 1 illustrates the nitrile group (-CN) was changed to amidoxime group (-C(NH_2_)=NOH). The treatment of the grafted films by addition of 3% hydroxyl amine (HO-NH_2_) in methanol–water mixture (1:1) and addition of KOH until pH 7 at 80 °C for 2 h. So, these resulted films were used as cation exchange membranes for heavy metals’ removal. The amidoxime group captures metal ions to form chelate compound. The chelate model structure is that one heavy metal ion is captured by two amidoxime groups. Therefore, the amidoxime absorbs selectively heavy metal ions [21].

### 2.4. Membrane Swelling Determination

The known weight of the treated and grafted membrane was submerged in distilled water for varying degrees of grafting, withdrawn from the water at various intervals for up to 24 h, then packaged using absorbent paper to catch any surface droplets, and weighed. The value of water uptake was calculated for each degree of grafting by using the following equation [22].
$$\mathrm{Water}\;\mathrm{uptake}\;\left(\%\right)=\frac{W_{s}-W_{g}}{W_{g}}\times 100$$
where W_g_ and W_s_ represent the weights of dry and wet membranes, respectively.

### 2.5. Applications (Metal Ion Adsorption by Amidoximated LDPE Film)

The generated membranes were examined for Pb (II) ion sorption using 0.1 gm of the grafted membrane and 20 mL of a solution of the ions containing 1:5 ppm in water (referred to by V). The sorption% was estimated at different time intervals using the following equation:$$\mathrm{Sorption}\;\left(\%\right)=\frac{A_{i}-A}{A}\times 100$$
where A_i_ and A are the initial and final concentrations in mg/L in the solution.

The total sorption capacity (the metal ion uptake capacity) of the synthesized copolymer for lead cations was calculated using the following formula:Q = V (C_1_ − C_2_)/W
where Q is the adsorption amount (mg/g of adsorbent), W the weight of the amidoximated PE film (g), V the volume of solution (L), and C_1_ and C_2_ are the concentrations (mg/L) of metal ions before and after adsorption, respectively.

## 3. Instrument and Apparatus

### 3.1. Gamma Radiation Source

The samples were irradiated at a dosage rate of (4.46–4.66) KGy/h using the CO^60^ Russian irradiation facility Gamma ray. The irradiation facility was built by Egypt’s Atomic Energy Authority, the National Center for Radiation Research and Technology (NCCRT).

### 3.2. Fourier Transform Infra-Red Spectroscopic Measurements

For both the original and grafted films, FTIR spectra were obtained in the range of 400–4000 cm^−1^ with the FT-IR 6300, Jasco, Japan.

### 3.3. Mechanical Measurements

The dumpel-shaped samples were 50 mm long, with a neck of 25 mm and 4 mm wide. The measurements of the tensile strength (Tb) and elongation percentage (Eb) were carried out using an Instron (Model-1195, High Wycombe, England) and the crosshead speed was 5 cm/min.

### 3.4. X-ray Diffraction

X-ray diffraction (XRD) measurements were done on blanks and produced polymers at room temperature on the Philips Pw 1730. The XRD pattern was recorded in the diffraction angle 2θ range. A scintillation counter was built into the X-ray generator. A nickel filter was used to create the diffraction patterns (CuKα) λ = 1.54 Å. The following experimental condition was used to obtain the X-ray diffractogram: filament current = 28 mA, voltage = 40 kV, and scanning speed = 20 mm/min.
D = Kλ cos θ/B_1/2_
where D is the particle size, K 0.89 is the Scherer constant linked to the crystal form, and the index (CuKα) λ = 1.54 Å is the X-ray wavelength, the diffraction angle, and the full width at half-maximum (FWHM in radian) [23].

### 3.5. Scanning Electron Microscopy

The surface morphology of the membranes was examined using a JEOL-JSM-5400 (Akishima City, Japan) Scanning Electron Microscope.

### 3.6. Thermal Gravimetric Analysis

For the thermal gravimetric analysis measurements, a model (TGA-50 Shimadzu, Thermo Gravimetric Analyzer made in Japan) was used (TGA). To avoid the oxidation of the polymer sample in this study, the nitrogen flow was kept constant at about 20 mL/min and the heating rate was 10 °C/min up to 600 °C.

### 3.7. Atomic Absorption Spectrophotometer (AAS)

This instrument is used for an elemental analysis approach that is effective for identifying trace metals in liquids, is essentially unaffected by the metal’s molecular structure in the sample and is carried out in the presence of other elements by the absorption of radiant energy by atoms at certain wave lengths characteristic for each one, resulting in transitions from a lower to a higher energy state; this depends on the concentration of the metal. The A-A scan-4 was used to analyze (AAS).

## 4. Results and Discussions

### 4.1. Effect of Solvent and Air Atmosphere

The effect of the solvent during the grafting process may change the grafting process by causing swelling in the polymer substrate. As a result, enhanced monomer accessibility and diffusion to the active site was produced by irradiation. However, such solvents also minimized the impact of radiation on the polymer and were sometimes referred to as protectors. Table 1 shows the effect of various solvent such as H_2_O, methanol, benzene, and a mixture of Methanol/H_2_O on the graft polymerization of AN/AAc comonomers onto PE films in a nitrogen atmosphere. The results showed that the dilution of the monomer’s mixtures used with methanol/H_2_O (30:70 wt%) leads to a remarkable increase in the grafting yield and reduces the homopolymerization process. This may refer to methanol/water as a polar system which shows a remarkable complete dissolution of the inhibitor, causing low homopolymerization. In addition, Table 1 shows that the air atmosphere may have an effect on the degree of grafting of low-density polyethylene (LDPE G%) and lower the rate of the reaction. This may be due to oxygen atoms reacting readily with the free radicals created, causing a lowering in the reaction yield of the degree of grafting, meaning bubbling in nitrogen increases the grafting yield.

#### 4.1.1. Effect of Inhibitor

As a radical scavenger, an inhibitor is one that prevents the formation of excess free radicals in each monomer during the initiation phase. This prevents homopolymerization when the concentration of the inhibitor is optimal. Table 2 suggested that with the enhancement of the grafting process in the presence of Fecl_3_ as an inhibitor, we can see that the addition of 2.5% by wt% ferric chloride effectively inhibits homopolymerization, and the grafting of the aqueous solution was significantly enhanced, followed by an effective decrease at a higher concentration. This may be due to the greater diffusivity of the comonomers solution into the bulk polymer. Thus, a higher grafting yield was achieved and less homopolymer formed in such a case, followed by the prevention of the graft process by the migration of the individual ions onto the substrate surface and blocking the active points. These can be considered as targets for copolymerization, as well as the inhibition of homopolymer formation.

#### 4.1.2. Effect of Irradiation Dose

In the present study, an irradiation technique was applied for a test tube containing AN/AAc solution with an inhibitor and LDPE film for the grafting process. It is well known that as the irradiation dose is increased, the concentration of free radicals produced increases, so increasing the grafting percentage. The grafting process or crosslinking creation occurs either in the polymer substrate or in the monomer solution. Figure 1 illustrates the relationship between the grafting % and radiation dose, and it also demonstrates that the yield of the grafting (AN/AAc) monomer mixture onto LPDE increases linearly with the radiation dose. In addition, the grafting yield increased with the dose to reach a maximum at 25 KGy, then it gradually decreased as the dose increased, and tended to level off at a dose higher than 25 KGy. This is because greater dosages lead to the development of a homopolymer, which restricts the diffusion of (AN/AAc) monomers through LDPE films. Radicals (active sites) are generated more often with the dose, but level out at higher doses due to their recombination. [24,25].

#### 4.1.3. Effect of Comonomers’ Composition

Some attention on enhancing grafting efficiencies is on the use of a mixed monomer system that leads to a more efficient grafting process from Figure 2a. It was observed that as the grafting yield increased, the (AN) content increased in the feed solution, which appeared most at point 6, corresponding to (80/20) (AN/AAc). It was observed that as the amount of AN increased in the comonomer composition, the degree of grafting increased to a maximum value (80/20). This may be due to the reactivity of the AN being higher than that of AAc. Additionally, the grafting process of AN/AAc comonomers may be enhanced in the presence of AN due to its higher polarity strength than AAc. These results suggest that in the comonomer composition, one of the binary monomers may enhance the grafting process, while the other monomer proceeds, and vice versa (synergistic impact). The grafting depends on the morphology of the films [26].

#### 4.1.4. Effect of Comonomers’ Concentration

For the grafting copolymerization process, the dilution impact of (AN/AAc) the binary monomer mixture with a composition of (80/20) wt% for LDPE is shown in Figure 2b. It has been demonstrated that as the comonomer concentration is increased, the degree of graft increases up to 75%, and after that, it decreases sharply. In several grafting systems, a maximum grafting yield has been recorded at a specific comonomer concentration. This has been linked to the grafting medium’s viscosity increasing when the monomer is transformed to a polymer and the propagation process is impeded by more than a mutual termination between two developing macro radical chains [27].

### 4.2. Characterizations of Grafted Membranes

#### 4.2.1. Swelling Behavior

Figure 3 illustrates the relationship between membrane water absorption and the degree of grafting of LDPPE-g-p-(AN/AAc). From the figures, it is seen that water uptake in terms of weight rise as the degree of grafting is increased to reach a maximum water uptake% of 79% for grafted films at 350%. It is well known that the swelling behavior of polyacrylonitrile is poor due to its hydrophobic nitrile group (-CN), which was changed to an amidoxime group (-C(NH_2_)=NOH) by a reaction with hydroxylamine (HO-NH_2_) after graft copolymerization. So, a part of the grafted copolymer membrane is converted to an absorbent agent by a chemical reaction. Figure 3 showed the significant increase in the water uptake for the treated membranes. This may be due to the conversion of the nitrile groups to the amidoxime groups that improves the swelling behavior of the grafted films. These findings show that the number of hydrophilic groups in the films determines the degree of swelling, i.e., the degree of grafting as well as the electrolyte’s shape. These findings are consistent with those reported in prior research [28,29].

#### 4.2.2. Infrared Spectroscopy (FTIR)

The IR spectra of LDPE blank, grafted, and amidoxaminated membranes are shown in Figure 4a–c respectively. In Figure 4a the characteristic bands of LDPE appeared around 2800, 1469, and 730 cm^−1^. The absorption peaks at 2800 and 1469 cm^−1^ are assigned to the CH_2_ group stretching and bending of a normal alkane, respectively. The last strong peak at 730 cm^−1^ is assigned to methylene CH_2_ rocking. The characteristic peaks for grafted LDPE films are shown in Figure 4c and 4d, respectively. The peak appeared around 3200–3500 cm^−1^, indicating the presence of a NH_2_ group of AN moieties. The intensity of this peak increases as the degree of grafting increases and, therefore, it is decreased slightly and did not disappear completely. This is due to the membrane that did not swell enough in the aqueous media before amidoxamination, and then, after the reaction began to be better, it efficiently swelled well with the conversion due to the replacement of the hydrophobic nitrile group (C=N) with the hydrophilic amidoxime group N=OH, which appears on the spectra at a broadband of 3200–3500 in Figure 4c. Moreover, the absorption band at 1660 cm^−1^ indicates the presence of the carbonyl group C=O of AN and AAc comonomers.

#### 4.2.3. Scanning Electron Microscope (Morphology)

Figure 5a shows the morphologies observed by SEM of an LDPE blank with a dense continuous smooth phase, such as most semi crystalline polymers. Figure 5b presents different types of surface textures in the case of LDPPE-g-p-(AN/AAc). From these figures, spots of grafted parts with different sizes can be seen. This indicates that the grafting of AN/AAc onto LDPE is not completely homogeneous as the distribution and concentration of the graft on the surface are not the same. In addition to Figure 5c for the amidoximated, grafted LDPE, the surface of the amidoxaminated polymer surface also shows that when there is no homogeneity in the distribution with linked grooves of varying widths, the surface becomes heterogeneous. This can increase the swell ability of the original membranes and cause the ions to adsorb inside their structure [30].

#### 4.2.4. X-ray Diffraction (XRD)

The technique of X-ray diffraction is used to determine the crystal chemistry, as the properties of known substances are never fully understood until the structure is known. To further understand the changes in the morphological structure generated by grafted polymeric materials, the XRD method was applied. Figure 6a–c shows the XRD diffractogram pattern for LDPE blank, LDPE-g-p(AN/AAc), and amidoxaminated membranes, respectively. In the case of blank and grafted LDPE films, Figure 6a–c shows that the radiation grafting of LDPE brought a drop in the degree of ordering of the polymeric material. These findings were evidenced from the observed drop in the relative intensity of the main diffraction line, and also due to its broadening. The radiation grafting process has a direct impact on the particles size of the polymeric materials, where the particles’ size of the grafted film increases as the degree of grafting increases.

The reduction in the crystallinity of the grafted LDPE and treated film may be noted, and that is due to the disordered structure implying the amorphous structure of the grafted comonomer components. These findings imply that the graft copolymerization of the comonomers’ (AN/AAc) binary combination has no effect on the crystallinity of the backbone polymers. The grafting occurs only in the amorphous areas, therefore, decreasing the overall crystallinity, which is observed also for the amidoxaminated films. This is due to the amorphous amidoxime group being grafted onto the grafted mixture, which dilutes the crystalline portion [31].

#### 4.2.5. Differential Scanning Calorimeter (DSC)

It is crucial to understand how the thermal characteristics and crystallinity of the LDPE-g-p(AN/AAc) system change, as well as how to characterize and determine chemical and physical changes. As diffusion is confined to the amorphous areas of polymers, the applications that rely on diffusion features of films, such as separation processes, need careful control of the crystallinity. The differential scanning colorimeter (DSC) is used to measure changes in the temperature characteristics of the produced membranes, as illustrated in Figure 7. The crystalline melting peak temperature for the grafted and treated film is somewhat changed to low temperatures, as seen in the figure. This change means that grafting makes changes in the crystalline regions due to the disordering and cross-linking occurring by grafting onto LDPE and the grafted membranes which are responsible for chain mobility restriction.

#### 4.2.6. Thermal Gravimetric Analysis (TGA)

The structure of the engrafted LDPE, grafted LDPE-g-p(AN/AAc), and amidoxaminated films were also characterized with the thermal analysis method in Figure 8a–c. The initial TGA thermograms were presented using a heating rate of 10 °C/min and a range of 100–500 °C. Analysis of the TGA data of the blank LDPE has one main degradation step that occurs between 200 and 600 °C for the blank LDPE film shown in Figure 8a, with a maximum rate of thermal decomposition at 450 °C [32].

In the case of the grafted LDPE, three distinct steps of weight loss were observed. The first step of weight loss in the range of 100–200 °C in Figure 8a may be attributed to the elimination of adsorbed moisture. There are differences in the weight loss behavior in the range of 100–200 °C in Figure 8b,c, which is probably due to the difference in the AN ratio where, with the increase of AN content in the composition, the hydrophilicity increases and thus the increase in weight loss occurs because of the elimination of adsorbed moisture. The second step of the weight loss observed by a smooth decrease in weight occurred up to 450 °C, which is due to the elimination of graft side chains. The latter decomposition step (third step), observed in temperatures above 500 °C, corresponds to the region of major weight loss occurring because of the extensive degradation of the polymer backbone chain leaving a residue. The increment in temperature weight loss may be due to the sequence distribution of the comonomer AN/AAc in the graft copolymer, which affects the thermal behavior as well as the nature of the comonomer at the decomposition temperature of 450 °C, at which the degradation reaction of the grafted side chains is completed.

These are the regions of major weight loss because of the extensive degradation of the polymer backbone chain leaving a residue (char) behind the final decomposition temperature (FDT). It was found that the residue (char yield) and the FDT values for both the grafted LDPEs increase with an increasing grafting degree. This is due to the increase in the amount of grafted P(AN/AAc) chains. This refers to the increase of the thermal stability upon grafting and the thermally stabilized CN group over the COOH group.

Thus, the analysis of the TGA curves of the original and grafted investigated polymers showed that the thermal stability of the grafted films increases in two ways: as the degree of grafting increases and by increasing the content of AN in the sample [33,34].

#### 4.2.7. Mechanical Measurements

In practical applications, physical attributes are quite important. The relationship between tensile strength and elongation percentage at the break and the degree of grafting was investigated in Figure 9a,b for the grafted membranes. The tensile strength improves steadily with the degree of grafting, as can be shown. However, as the degree of grafting rises, the elongation percentage falls. The rigidity of the prepared membranes generally increases as the degree of grafting increases. As a result, a reduction in elongation is envisaged. The tensile strength of the graft copolymer increases as the acrylonitrile percentage rises. It is critical that the grafted films have a high tensile strength, especially if they are going to be used for ion-exchange separation. The developed grafted membranes have good tensile qualities, making them suitable for use in practical applications where thermal parameters are changed by grafting.

### 4.3. Application on Sorption Behavior of Pb (II)

#### 4.3.1. The Mechanism of Ion Sorption

The grafting of various vinyl monomers onto the polymers by graft copolymerization will directly affect the adsorption capacity of different metal ions by the newly improved membranes, and this is due to the existence of the new function groups which make a significant change in the membrane surface. This increase in the adsorption of the reactive grafted membranes can be attributed to the existence of multifunctional groups such as carbonyl and amide groups from the AN and AAc monomers, respectively, which are grafted onto the original membranes [33].

#### 4.3.2. Effect of Degree of Grafting

Figure 10a depicts the influence of grafting yield on Pb (II) chelation through LDPE-g-p-(AN-AAc) membranes. For all membranes studied, it is evident that the amount of metal ion uptake increases as the degree of grafting increases. This clearly shows that the number of functional reactive groups in the LDPE-g-p-(AN-AAc) membranes is the most important factor in metal chelation or complexation [35].

#### 4.3.3. Kinetics of Sorption

For Pb removal, the effect of contact time on the rate of the maximum metal uptake or sorption from a neutral solution with a concentration of 5 ppm was studied and shown in Figure 10b. This graph illustrates metal sorption as a function of contact or treatment time using a cationic-treated membrane. It is clear from the figures that with time, the metal uptake rises until it reaches its highest point (it is referred to as the maximum membrane capacity) after 24 h for all metals investigated. Metal sorption across porous ionic membranes is dependent on different factors such as the polarity of the metal ion, its electronic structure and ionic radius, as well as the type of its interaction with the membranes’ functional groups [36].

#### 4.3.4. Influence of pH on Sorption

The pH of the solution affects the overall surface charge of the adsorbent LDPE-g-p-(AN-AAc) and the charge on the adsorbate Pb (II). The results of pH variation showed that the maximum metal removal was found at pH 9 with an accumulation of 80%, as shown in Figure 10c.

At pH 9, the adsorption of Pb (II) reaches the optimum, which may be due to the increase of OH^-^ (hydroxyl ions), but slightly decreases until pH 10, as shown in Figure 10c. Then, a major drop is observed at pH > 10. This might be attributed to the precipitation of Pb (II) at pH ∼ 9.4. Hence, the sudden uptake of Pb (II) by LDPE-g-p-(AN-AAc) at pH < 7 is not attributed to the formation of Pb(OH)_2_. The hydrogen ion competes with accessible exchange sites at low pH levels, reducing the absorption of Pb cations. The H+ decreases as the pH rises, and the proton’s competitive action on the metal cation decreases, resulting in an increase in the metal sorbet percentage of Pb(II) ions [8].

The mechanism of pH dependence of Pb(II) ions’ uptake can be explained by the nature of composite surface–metal binding sites based on the study of Pb (II) adsorption by mixed fly ash [37], which pointed out that heavy metal ion adsorption could be divided into stages, precipitation and hydrolyses [38]. The ability of LDPE-g-p-(AN-AAc) to adsorb Pb(II) is attributed to the presence of ions, such as OH− and NH^2+^ groups, and the highly negative surface of LDPE-g-p-(AN-AAc), which could attract positively charged (M^+^) [39,40]. In conclusion, the results in Figure 10c, indicate that the optimum pH values of the binary system of the polymer blends, to remove Pb (II) from the solution by using LDPE-g-p-(AN-AAc), are 8–10.

#### 4.3.5. Effect of Metal Ion Concentration

Analyses into the sorption capacity of the generated amidoxaminated membranes towards Pb ions were carried out (Table 3 and Table 4). The effect of changing the metal ion concentration on the sorption of Pb by LDPE-g-p-(AN-AAc) was investigated. The data optioned are shown in Figure 10d and illustrate that as the concentration of metal ions rises, the sorption also increases until a certain concentration (3 ppm), and after that, it decreases. This may be attributed to the active sites (amidoxime and hydroxyl function groups), which are responsible for chelation, being saturated with the metal ion at certain concentrations and, at very higher concentrations, there is no available space for adsorption, which may lead to the decrease in the sorption percentage [41,42].

Total sorption capacity for lead ions on membranes (Table 3) can be calculated by equation
Q = V (C_1_ − C_2_)/W

## 5. Conclusions

The preparation of LDPE-g-p-(AN-AAc) films was synthesized using the pre-irradiation technique. A mixture of 30% methanol and 70% H_2_O was used as a solvent in nitrogen, and 2.5% by wt% FeCl_3_ as an inhibitor was found to prevent homopolymerization. The highest graft yield obtained was 320% at a 25 kGy radiation dose, 80/20 monomer composition, and 60% comonomer concentration. The AN/AAc-grafted films were modified with hydroxylamine hydrochloride to prepare an amidoxime adsorbent. The prepared adsorbent was characterized by the water uptake %, FTIR, SEM, XRD, DSC, TGA, and mechanical properties. The prepared amidoxime adsorbent showed a high affinity towards Pb (II) adsorption. The highest adsorption capacity obtained was 150 mg/g after 50 h contact time at pH 9 and an initial metal ion concentration of 5 ppm.

## Data Availability

Not applicable.

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
