# Peer review of "Synthesis of Amidoxime Adsorbent by Radiation-Induced Grafting of Acrylonitrile/Acrylic Acid on Polyethylene Film and Its Application in Pb Removal"

_polymers, 2022, doi:10.3390/polym14153136_

Round 1

Reviewer 1 Report

This study reports the appropriate settings for the grafting technique in terms of maximizing grafting yield and improving the characteristics of the grafting films produced. The impact of solvent, inhibitor concentrations, monomeric mixture composition, and irradiation dosage were studied systematically. The grafted acrylonitrile/ Acrylic acid on Polyethylene Film was characterized by FTIR, SEM, DSC, TGA, and XRD. The application of such films for the maximum sorption of Pb ions as water contaminants under different conditions of degree of grafting, contact time, metal ion concentration and pH were investigated. The results obtained suggest that the fabricated material can be used as adsorbent

Overall, this article is written nicely with some important results. However, there are some issues with respect to novelty and literature review section and also some formatting issues needs to be addressed before final acceptance. Authors are suggested to revise the manuscript carefully based on the points mentioned below.

  • Important results are missing in abstract
  • The novelty part of the manuscript should be highlighted in better way. Why grafting of Acrylonitrile/ Acrylic acid on Polyethylene Film was chosen for this study? How it is better than only Acrylonitrile or only Polyethylene Film?
  • Please strengthen the literature review section with some background information of adsorption technology in wastewater treatment with reference to recent works. Following articles may become useful to authors.   https://doi.org/10.1007/s00289-021-03546-8; https://doi.org/10.1080/01932691.2020.1845958; https://doi.org/10.1016/j.carbpol.2020.117564; https://doi.org/10.1007/978-981-19-1516-1_2; https://doi.org/10.1016/j.jhazmat.2021.125696; https://doi.org/10.1016/j.enmm.2021.100506

  • Please add the error bars in the Fig and Tables, wherever applicable.
  • Please explain the typical behavior of Fig 4 in details with appropriate reference.
  • Fig 7, X and Y axis text are not readable. Need formatting works.
  • Fig. 9, poor resolution. Need to improve
  • Typical behavior shown in TGA plot should be discussed in details with appropriate references.
  • The results obtained in this study should be compared with the similar works for Pb ion removal to evaluate the potential of this work.
  • Too many Figs, total 16 nos. Authors may combine few figs to reduce the total fig number within ten.

Author Response

  • Point 1. The novelty part of the manuscript should be highlighted in better way. Why grafting of Acrylonitrile/ Acrylic acid on Polyethylene Film was chosen for this study? How it is better than only Acrylonitrile or only Polyethylene Film?

Response 1: the chosen for binary mixture of AN/AAc to increase the efficiency of removal by increasing the number of functional gorups on the grafted membranes

  • Point 2: Please strengthen the literature review section with some background information of adsorption technology in wastewater treatment with reference to recent works. Following articles may become useful to authors.   https://doi.org/10.1007/s00289-021-03546-8; https://doi.org/10.1080/01932691.2020.1845958; https://doi.org/10.1016/j.carbpol.2020.117564; https://doi.org/10.1007/978-981-19-1516-1_2; https://doi.org/10.1016/j.jhazmat.2021.125696; https://doi.org/10.1016/j.enmm.2021.100506

Response 2: Please provide your response for Point 2. (in red)

  • Point 3: Please add the error bars in the Fig and Tables, wherever applicable.

Response 3: Please provide your response for Point 3. (in red)

  • Point 4 Please explain the typical behavior of Fig 4 in details with appropriate reference.

Response 4: Please provide your response for Point 4. (in red)

  • Point 5: Fig 7, X and Y axis text are not readable. Need formatting works.

Response 5: Please provide your response for Point 5. (in red)

  • Point 6 : Fig. 9, poor resolution. Need to improve

Response 6: Please provide your response for Point 6. (in red)

  • Point 7 : Typical behavior shown in TGA plot should be discussed in detail with appropriate references.

Response 7: Please provide your response for Point 7. (in red)

  • Point 8 : The results obtained in this study should be compared with the similar works for Pb ion removal to evaluate the potential of this work.
  • Response 8: Please provide your response for Point 8. (in red).

Reviewer 2 Report

The manuscript by R.F. Khedr named “Synthesis of Amidoxime Adsorbent by Radiation Induced Grafting of Acrylonitrile/ Acrylic acid on Polyethylene Film and Its Application in Pb Removal” is devoted to the synthesis of LDPE and polyacrylonitrile and acrylic acid graft-copolymers, research of the effect of synthesis conditions on grafting efficiency, and investigation of lead ions sorption by the synthesized copolymers. Synthesized copolymers were characterized by some methods, such as FTIR, SEM, XRD, DSC, and TGA and it was shown a possibility to use obtained copolymer as metal ion sorption membrane. The work has a typical structure and a lot of experimental data. However, due to the poor English and figure resolution, it is very hard to understand what the author wants to represent and explain. For a qualitative and complete review of this work, first of all, it is necessary to completely proofread English, as well as provide high-resolution drawings, especially FTIR spectra, X-ray patterns, and sorption curves. Also, the manuscript is sloppy: many sentences are missing periods in the end or they do not start with a capital letter. All this makes it very difficult to perceive information and requires serious processing for a qualitative review.

In addition, there are several other remarks:

1. In the Materials and reagents subsection, there is no information on the purity grade of the reagents used, whether they were additionally purified, etc. In the work, the author evaluates the swelling properties of graft copolymers, and the purity of the monomers has a significant impact on this characteristic.

2. The given in the Material and Method section equations are unreadable, so one can only guess how the calculations were made. This needs to be edited.

3. In paragraph 3.4, the author indicates in line 136 the excitation wavelength of 1.45 angstroms, and in line 136 the value of 1.54 angstroms is already present. If we are talking about a copper emitter, then 1.54 is the correct value.

4. The author used an inhibitor of iron (III) chloride in the work, it was shown that the most effective grafting occurs when its content is 2.5% wt. This is fairly high content, but the manuscript does not contain information about how the resulting polymer was purified from iron salts or about their content in it.

5. On the Y-axis in Fig. 6 shows the water content, but it is more correct to reflect the swelling ratio along this axis. In addition, in the Materials and Methods section, the author provides an equation for calculating this parameter. Most likely for the swelling ratio. The equation is not readable, so I can only guess...

6. The author begins the discussion of the experimental results with a synthesis condition that influences the grafting efficiency. And only after that evidence of grafting in the form of IR spectra is given. This is illogical. First, we must prove that we have a structure, and then discuss the influence of various parameters on the process of its formation. It is necessary to represent spectra in high resolution, they are not readable. Also, expand their description, and reflect the absorption bands of all grafted monomers and signs of LDPE modification. In addition, it would be good to determine the compositions of the resulting copolymers.

6. XRD patterns are also unreadable. The signal description is missing. If their representation is only to show a decrease in the intensity of the modified LDPE signals compared to the initial one, then it is better to present all three X-ray patterns in one figure. In high definition, of course.

7. The author provides DSC data to assess the change in the crystallinity of the LDPE structure after grafting. From DSC data, it can find the glass transition temperature, changes in the values of which will perfectly demonstrate changes in the structure of polymers.

8. Table 3 should be deleted. It duplicates the data in Figure 11 and its discussion.

9. And more about the drawings. It is necessary to make the labels to the axes uniform. This is especially true for the drawings devoted to the sorption properties of the resulting polymers.

10. The name of subsection 4.3.2 should be changed to Kinetics of sorption. And what about the mechanism of ion sorption?

11. Title of section 4.3.3. should be changed to Influence of pH on sorption. Also, the changes in the graft copolymer structure at different pH, as well as features of lead cation under the same conditions should be discussed in more detail.

12. What is the total sorption capacity of the synthesized copolymer for lead cations?

Based on mentioned above, I think that manuscript should be reconsidered after very major revision.

Author Response

Point 1. In the Materials and reagents subsection, there is no information on the purity grade of the reagents used, whether they were additionally purified, etc. In the work, the author evaluates the swelling properties of graft copolymers, and the purity of the monomers has a significant impact on this characteristic.

Response 1: Please provide your response for Point 1. (in red)

Point 2: The given in the Material and Method section equations are unreadable, so one can only guess how the calculations were made. This needs to be edited..

Response 2: Please provide your response for Point 2. (in red)

Point 3: In paragraph 3.4, the author indicates in line 136 the excitation wavelength of 1.45 angstroms, and in line 136 the value of 1.54 angstroms is already present. If we are talking about a copper emitter, then 1.54 is the correct value.

Response 3: Please provide your response for Point 3. (in red)

Point 4: The author used an inhibitor of iron (III) chloride in the work, it was shown that the most effective grafting occurs when its content is 2.5% wt. This is fairly high content, but the manuscript does not contain information about how the resulting polymer was purified from iron salts or about their content in it.

Response 4: Please provide your response for Point 4. (in red)

Point 5: On the Y-axis in Fig. 6 shows the water content, but it is more correct to reflect the swelling ratio along this axis. In addition, in the Materials and Methods section, the author provides an equation for calculating this parameter. Most likely for the swelling ratio. The equation is not readable, so I can only guess.

Response 5: Please provide your response for Point 5. (in red)

Point 6 :The author begins the discussion of the experimental results with a synthesis condition that influences the grafting efficiency. And only after that evidence of grafting in the form of IR spectra is given. This is illogical. First, we must prove that we have a structure, and then discuss the influence of various parameters on the process of its formation. It is necessary to represent spectra in high resolution, they are not readable. Also, expand their description, and reflect the absorption bands of all grafted monomers and signs of LDPE modification. In addition, it would be good to determine the compositions of the resulting copolymers.  XRD patterns are also unreadable. The signal description is missing. If their representation is only to show a decrease in the intensity of the modified LDPE signals compared to the initial one, then it is better to present all three X-ray patterns in one figure. In high definition, of course.

Response 6: Please provide your response for Point 6. (in red)

Point 7 : XRD patterns are also unreadable. The signal description is missing. If their representation is only to show a decrease in the intensity of the modified LDPE signals compared to the initial one, then it is better to present all three X-ray patterns in one figure. In high definition, of course.

Response 7: Please provide your response for Point 7. (in red)

Point 8 :Table 3 should be deleted. It duplicates the data in Figure 11 and its discussion.

Response 8: Done

Point 9 : And more about the drawings. It is necessary to make the labels to the axes uniform. This is especially true for the drawings devoted to the sorption properties of the resulting polymers.

Response 9: Please provide your response for Point 9. (in red)

Point 10: The name of subsection 4.3.2 should be changed to Kinetics of sorption. And what about the mechanism of ion sorption?

Response 10: Please provide your response for Point 10 (in red)

Point 11:Title of section 4.3.3. should be changed to Influence of pH on sorption. Also, the changes in the graft copolymer structure at different pH, as well as features of lead cation under the same conditions should be discussed in more detail.

Response 11: Please provide your response for Point 11 (in red)

Point 12 :What is the total sorption capacity of the synthesized copolymer for lead cations?

Response 12: Please provide your response for Point 12 (in red)

Please see the attachment."

Round 2

Reviewer 2 Report

The author made required changes. The scientific part of this manuscript is correct, adequate and possesses a potential interest for Polymers readers. However, English must be corrected, as well as figures must be resubmited in higher resolution. To my mind, manuscript may be accepted after correction of these drawbacks.

Author Response

English must be corrected, as well as figures must be resubmited in higher resolution. To my mind, manuscript may be accepted after correction of these drawbacks.

Reply 
